# Who Cares for Visiting Nurses? Workplace Violence against Home Visiting Nurses from Public Health Centers in Korea

**DOI:** 10.3390/ijerph17124222

**Published:** 2020-06-13

**Authors:** Eunjoo Kim, Heeseung Choi, Ju Young Yoon

**Affiliations:** 1College of Nursing, Seoul National University, Seoul 03080, Korea; kookie1@snu.ac.kr (E.K.); hchoi20@snu.ac.kr (H.C.); 2Research Institute of Nursing Science, Seoul National University, Seoul 03080, Korea

**Keywords:** visiting nurses, workplace violence, home visits, risk factors, safety management

## Abstract

Visiting nurses are commonly exposed to workplace violence, but there is a lack of research on violence against these nurses. The purpose of this study was to identify visiting nurses’ workplace violence experiences during home visits. This study used a mixed method design. Survey data of 357 home visiting nurses from public health centers were collected for the quantitative data, and a focus group interview was conducted with six visiting nurses for the qualitative data. The quantitative data were analyzed using logistic regression, and the qualitative data were analyzed using content analysis. Younger, temporary visiting nurses and those who had previously been exposed to violent clients had a higher risk of workplace violence. The violence visiting nurses faced included not only violence during the visits but also unpredicted danger and harassment after the visit. After experiencing a violent event, visiting nurses’ attitudes and emotions changed toward nursing services. Visiting nurses were likely to deal with violence at the individual level given the insufficient organizational support system. An organizational-level safety management system should be established based on the characteristics of workplace violence risks and the nurses’ experiences in this study.

## 1. Introduction

The importance of community-based health care is being highlighted worldwide to manage the growing elderly population and chronic conditions. Against this backdrop, several countries, including the Republic of Korea (hereafter, “Korea”), have expanded home healthcare services in public health and long-term care settings [1,2]. Visiting health professionals have a wide range of occupations including nurses, psychiatrists, and social workers [3]. However, the majority of the home-visiting staff are registered nurses (RNs) in Korea [4]. Although there are differences in organizational characteristics across countries depending on the healthcare system, the working environment in which health professionals visit patients’ homes and provide health care services is very similar.

Visiting nursing is one of the most fundamental and effective nursing services for community-dwelling persons because it provides customized health services. Paradoxically, there is a crucial difference in worker safety between home visits that provide tailored services compared to general work situations [5] because the patient’s home is a space beyond the protection of the organization. Many visiting nurses worldwide have reported that they have been exposed to workplace violence in home settings [4,5,6,7,8,9].

The International Labour Organization (ILO) defines workplace violence as “any action, incident or behavior that departs from reasonable conduct in which a person is assaulted, threatened, harmed, injured in the course of, or as a direct result of his or her work” [10]. The University of Iowa Injury Prevention Research Center (UIIPRC) [11] classifies workplace violence into four categories according to the relationship between the perpetrator and the victim. A perpetrator is (a) an outsider with criminal intent (type I), (b) a customer or client (type II), (c) an employee or past employee (type III), or (d) a person with a personal relationship (type IV).

For visiting nurses, type II workplace violence occurs most commonly because the main task is to visit the person’s home to connect with and directly face the patient and the family members (hereafter, “client”) [7]. In previous studies, most workplace violence toward visiting nurses has been verbal violence. This is followed by threats and sexual violence [4,5,6,7,9]. Although the frequency is relatively low, physical assault clearly occurs and is reported [4,6,7,8,9]. In addition, health-related consequences (e.g., negative emotional or posttraumatic stress disorders) and job-related consequences (e.g., job satisfaction or shortening home care visits) have been reported [5,6,7,12,13].

Visiting personnel have recently raised concerns about safety issues. In response, some countries have introduced legal action. Relatively early, in 2007, Europe established the Framework Agreement on Harassment and Violence at Work [14]. The Workplace Violence Prevention for Health Care and Social Service Workers Act was proposed in the United States in 2018 [15]. In Korea, the Occupational Safety and Health Act (Law No. 15588) was revised in 2018, and Article 26-2, “Preventive measures against health problems caused by verbal abuse by customers”, was established [16]. This statute mandates that employers develop detailed policies to protect workers.

However, results from empirical research in the field of visiting nursing is insufficient. Studies have been conducted to identify workplace violence in the healthcare sector in general [17], but few studies have focused on visiting nurses [3]. Because the workplace of the client’s home is an unpredictable and uncertain environment, it is important to understand the risk factors that visiting nurses perceive [8]. In particular, it is important to confirm the perception of visiting nurses because the perceived risk factors are different between administration and staff [18]. Because the development of the risk assessment screening tool was based on the expert opinion, it was insufficient to consider the risk factors actually perceived by visiting nurses [19]. Although the most important characteristic of visiting nursing is face-to-face work, previous studies have not adequately considered the context, tasks, and work environment of visiting nurses in the workplace risk factors [12,18]. In the previous studies, workplace risks recognized by home visit staff are often identified only from the perspective of a criminal environment (e.g., guns, weapons, and gangs) [20,21], and rarely from the perspective of the impact of high-risk client exposure routinely encountered by visiting nurses [22]. In this study, information about these environmental aspects, especially working with high-risk clients, was sought in identifying workplace risk factors. Ten years ago, Hutchings and colleagues conducted a safety program as a proactive approach with three components: a risk assessment screening tool, a sign-in/sign-out system, and a buddy system [19,23]. Fujimoto et al. recently reported a low rate of implementation of preventive measures for the safety of visiting nurses [24]. However, an explanation of this result was insufficient due to the general limitations of survey research. In addition, previous focus group interviews with visiting nurses were outdated (e.g., pagers vs. smartphone) [25]. Thus, it is necessary to confirm to what extent the current organizational system contributes to the problem. It is unclear whether a sufficient safety management system is in place to ensure that visiting nurses find utility and are protected in the workplace. In particular, home visiting nurses mainly visit a vulnerable population in an uncontrolled environment (i.e., private homes) in Korea. Despite the high risk of workplace violence in this environment when compared to hospital nurses, the safety management system does not adequately support them. Thus, both research and policy development are needed [4]. This study focused on identifying leverage points for intervention based on visiting nurses’ reports of the current state of workplace violence.

The purpose of this study was to explore the home visiting nurses’ experiences of workplace violence, and to identify visiting nurses’ perceptions of the risk factors and organizational management of workplace violence. In this study, a mixed method research (MMR) approach was used to provide a comprehensive understanding of the workplace violence experience of visiting nurses through quantitative research and to delve deeper into the participants’ perspectives through qualitative research [26].

## 2. Materials and Methods

### 2.1. Study Design and Sample

This study used a mixed methods design to investigate workplace violence against home visiting nurses. Based on the explanatory sequential design among the mixed methods research classified by Creswell et al. [27], a quantitative survey was first conducted on the violent experiences of visiting nurses, which was followed by a qualitative study. In this study, the mixed methodology provides a thorough understanding of research issues through quantitative research (structured questionnaires) and then refines and explains the statistical results by exploring participants’ perspectives in more depth through qualitative research (focus group interview, FGI) [26].

Participants were home visiting nurses recruited from S city public health centers. The researcher visited the training site for visiting nurses in the S city Visiting Health Care Project, explained the purpose and procedure of the study to the visiting nurses, and recruited participants. Since the workplace violence experience measurement tool measured workplace violence within the previous year, the study participants were limited to those who had worked as a visiting nurse for more than one year. When recruiting survey participants, the recruiting flyers for the focus group interview were distributed together so that the visiting nurses who wish to participate in the focus group interview were able to directly contact the researchers. This was done to ensure that the participation of the focus group interview for study participants was not unnecessarily known to the researcher and to people other than the same interview group participants.

The survey questionnaires were distributed to 532 in-service nurses. The population was the total number of in-service visiting nurses with more than one year of experience in S city, excluding visiting nurses who have not participated in education due to sick leave or leave of absence. The researcher explained the purpose of this study and 368 nurses who agreed to participate submitted their questionnaires (response rate: 69.2%). Survey data from 357 nurses were included in the analysis after deleting 11 incomplete responses.

An appropriate number of participants in a focus group is 6–10, according to qualitative research guidelines [28]. It is difficult to maintain discussion if there are less than six participants, and difficult to control if there are more than 10 participants [28]. A focus group interview was conducted with six nurses (one group) who had completed the survey and had reported experience with workplace violence in the previous year and agreed to the interview.

### 2.2. Measures and Data Collection

Questionnaires included measures of workplace violence experience, perceived violence risk factors (individual risk factors, workplace risk factors), and the safety management system of their organization.

Workplace violence experience was measured based on the nurses’ reported exposure and experience of violence using the Korean Workplace Violence Scale (K-WVS) subscale developed by Chang et al. [29]. The instrument consists of 10 items to measure experience of psychological or sexual violence from a customer, experience of psychological or sexual violence from a supervisor or coworker, and experience of physical violence from a customer, supervisor, or coworker. In this study, only five items about the experience of violence from a customer were used by considering the definition of workplace violence in this study. Response options included never, sometimes, frequently, or very often, but responses were dichotomized by categorizing never responses as “no” and responses of sometimes, frequently, or very often as “yes”. Chang and colleagues [29] reported Cronbach’s alpha as 0.51–0.75. Cronbach’s alpha in the present study was 0.66.

As for the perceived violence risk factors, individual risk factors included the visiting nurses’ characteristics such as gender, age, work experience, and appearance. Appearance referred to wearing a nurse’s uniform or displaying an identification (ID) card when making a home visit. Workplace risk factors included working alone, hazardous residential environment, and working with high-risk clients. Working alone included a response option of 0–100% for the percentage of visits they conducted by themselves during the past year. For a hazardous residential environment and working with high-risk clients, we used the pre-visit safety assessment checklist items included in the Seoul Metropolitan Government’s Visit Safety Manual [30]. The following questions were asked with a response option of 0–100% corresponding to those who were in charge within the last year. The hazardous residential environment used the question: “Is the patient’s residence isolated (vulnerable homes such as cheap and narrow dorms, side rooms, underground or the like) or does the area have a high crime rate?” When working with high-risk clients, we used the question: “Do patients or their families have high risk factors (psychiatric disorders, history of suicide attempt, alcoholic, domestic violence assailant, child abuser, criminal record, sex offender, history of violence against visiting nurses, in distress)?“ Responses to high risk factors of clients were measured separately.

The safety management system was measured based on the organizational protective system for workplace violence, using the Korean Workplace Violence Scale (K-WVS) subscale developed by Chang et al. [29]. The instrument consists of 14 items. This study used only nine items about the experience of violence from customers because it was limited to workplace violence from visiting nursing clients. These nine items assessed both preventive measures of workplace violence and post-event management. After discussion with the original author who developed the instrument, these nine items were divided into two domains: preventive measures (5-items) and post-event management (4-items). Explorative Factor Analysis (EFA) was performed by applying the principal component extraction method with varimax rotation. This process resulted in a two-factor model with nine items. Factor I was labelled ‘preventive measure’ with five items with high loadings (0.84–0.65). Factor II was labelled ‘post-event management’ with four items with high loadings (0.82–0.76). Response options included strongly agree, agree, disagree, or strongly disagree. A yes/no dichotomized version of this variable was derived by categorizing strongly agree or agree responses as “yes” and responses of disagree or strongly disagree as “no”. Chang and colleagues [29] reported the Cronbach’s alpha as 0.97 and the Cronbach’s alphas in the present study were 0.87 (preventive measures) and 0.88 (post-event management).

We collected qualitative data using FGI. The focus group interview lasted two hours in a quiet seminar room in the researcher’s university. The principal investigator (PI) moderated the group interview, and a research assistant recorded important content during the group interview. The PI, who conducted the interview, encouraged visiting nurses to comfortably share opinions and feelings about workplace violence based on the PI’s experience of researching public health centers and operating visiting nurse training courses. The moderator and assistant debriefed the group immediately after the interview to ensure objectivity. The interview was recorded with the participants’ permission, and the recorded material was immediately transcribed on the day of the interview. During the interview, an interview guide (semi-structured questionnaire, Appendix A) was used to further investigate the research questions in accordance with the participant’s story flow. This study was conducted under the ethical approval of the Seoul National University Institutional Review Board (IRB No. 1905/002-012).

### 2.3. Data Analysis

We conducted descriptive analyses of the quantitative data. Logistic regression was conducted to identify factors by affecting the visiting nurses’ workplace violence experience. The multiple imputation (MI) function of SPSS software was used in the logistic regression to handle missing data [31]. All quantitative data were analyzed using the IBM SPSS 23.0 program.

After the interview transcript, we received feedback from the participants about the content. The qualitative data analysis was based on an inductive content analysis approach [32], which repeatedly classified the characteristics and frequencies found in the process of repeatedly reviewing qualitative data from the FGI. In addition, after data analysis, we asked the supervisor to confirm whether the results of the study sufficiently reflected her experience with visiting nurses’ workplace violence to achieve fittingness.

### 2.4. Research Team and Reflexivity

The research team operated visiting nurse training courses and conducted research regarding violence within the visiting nurses’ workplaces. The first author has been conducting research in workplace violence in service occupations based on the work experience in the airline’s aeromedical center. The corresponding author, as a professor of community health nursing, has been researching public health center personnel. The second author is a mental health nursing professor with many years of experience in qualitative and mixed research.

## 3. Results

### 3.1. Participant Characteristics

Participant characteristics are shown in Table 1. The survey respondents were mostly women (*n* = 354, 99.2%), which represents the demographics of this profession. The mean age was 50.15 years (SD = 6.75) and most participants were married (*n* = 335, 94.1%). As for the education level, more than half had an associate of arts degree in nursing. The overall work experience as an RN including as a visiting nurse averaged about 14 years, 7 months (174.46 ± 71.48 months) while the work experience as a visiting nurse averaged about 5 years, 9 months (68.46 ± 43.74 months). In terms of the type of employment, they were mostly permanent employees (*n* = 310, 87.6%). The average number of registered patients was 460.51 (SD = 202.03), and the average number of home visits per month was 81.51 (SD = 28.73). Only 73 nurses (20.4%) reported wearing uniforms during the home visits, but 255 nurses (71.4%) reported wearing ID cards.

### 3.2. Quantitative Results

#### 3.2.1. Workplace Violence Experience

Table 2 shows the results of the workplace violence experiences of visiting nurses. During the entire period of working as a visiting nurse, 270 visiting nurses (75.8%) reported experiencing workplace violence, while 240 (67.2%) visiting nurses had experienced workplace violence within the last year. The average Korean Workplace Violence Scale (K-WVS) score was 1.47 (SD = 1.41). By type of violence, 53.5% visiting nurses experienced verbal abuse, 30.3% experienced sexual violence, 28.0% experienced threats or harassment, 30.9% experienced discrimination, and 2.2% experienced physical assault. 

#### 3.2.2. Perceived Workplace Risk Factors

Table 3 shows the level of exposure to workplace risk factors perceived by visiting nurses within the last year. Hazardous residential conditions averaged 33.6% (SD = 27.2), and the rate of working alone averaged 62.1% (SD = 34.9). According to the survey of the ratio of face-to-face work for high-risk clients included the following: 17.3% with psychiatric disorders, 6.9% had attempted suicide, 12.1% were alcoholics, 5.3% had committed domestic violence, 2.3% had committed child abuse, 8.1% had a criminal record, and 3.3% were sex offenders. The results also indicated that 2.8% (SD = 6.7) of the nurses had experienced violence from the visited clients, and 25.5% (SD = 25.5) visited clients were in distress at the time of a visit.

#### 3.2.3. Safety Management System

The visiting nurses’ perceptions of the safety management systems of their organizations are shown in Table 4. Among the preventive measures, 73.3% of the visiting nurses said that there was no process to identify a violent event, and 67.4% thought that there was no device or system to protect them against violence. As for post-event management, 67.0% said that the organization did not address or solve the problem of violence from a client. In addition, 55.3% of the visiting nurses said they did not have a helpful supervisor when they were exposed to violence.

The average score of the preventive measure items among the factors of the organizational violence protection system of the K-WVS was 5.32 (SD = 3.61, range: 0–15). The average score of post-event management items was 5.35 (SD = 3.14, range: 0–12).

#### 3.2.4. Effect of Risk Factors and Preventive Measures on Workplace Violence Experiences

Table 5 shows the effects of risk factors and preventive measures on overall workplace violence experiences. As nurses’ age increased by one year, the probability of experiencing violence decreased by 9% (95% CI, 0.86 to 0.95). Temporary workers were 2.95 times (95% CI, 1.10 to 7.88) more likely to experience violence than permanent workers. Visiting nurses who were exposed to clients who had been violent against visiting nurses were 3.76 times (95% CI, 1.97 to 7.15) more likely to experience violence than those who were not.

In terms of the effects of the specific types of violence, the significant variables affecting the workplace violence experience were different. In the case of verbal violence, the statistically significant variables were age (OR = 0.91, 95% CI, 0.87 to 0.95), type of employment (OR = 2.66, 95% CI, 1.05 to 6.72), number of registered patients (OR = 1.71, 95% CI, 1.01 to 1.35), and whether or not they had been exposed to clients who had been violent against visiting nurses (OR = 4.97, 95% CI, 2.73 to 9.06). For sexual violence, the statistically significant variables were age (OR = 0.96, 95% CI, 0.91 to 1.00), exposure to clients who had been violent against visiting nurses (OR = 2.22, 95% CI, 1.26 to 3.91), and sex offender exposure (OR = 2.90, 95% CI, 1.49 to 5.65). For threats or harassment, the statistically significant variables were age (OR = 0.95, 95% CI, 0.90 to 0.99), type of employment (OR = 3.57, 95% CI, 1.44 to 8.81), and whether or not they had been exposed to clients who had been violent against visiting nurses (OR = 3.78, 95% CI, 2.06 to 6.92). For discrimination, the statistically significant variables were age (OR = 0.94, 95% CI, 0.90 to 0.99), type of employment (OR = 3.18, 95% CI, 1.30 to 7.78), exposure to a child abuser (OR = 2.75, 95% CI, 1.49 to 5.09), and whether or not they had been exposed to clients who had been violent against visiting nurses (OR = 1.91, 95% CI, 1.02 to 3.59).

The significant results are shown in Table 5. Appendix B shows the correlation between workplace violence experience and risk factors. Appendix C reports the overall logistic regression results. Physical assault was excluded from the logistic regression analysis due to too few events.

### 3.3. Qualitative Results

As a result of the FGI, two categories, four themes, and 16 subthemes were identified for the visiting nurses’ workplace violence experience (Table 6). The first category, which involved violence experiences and consequences, identified the characteristics of violence against visiting nurses and the changes in the visiting nurses after violent experiences. Another category, which involved reality of managing workplace violence, confirmed the reality of visiting nurses attempting to respond personally given the poor organizational management of their organization.

#### 3.3.1. Violence Visiting Nurses Faced

When visiting nurses visited their clients, direct physical assaults were uncommon, but verbal assaults were frequently reported. In particular, when clients were dissatisfied with the welfare benefits, they often expressed anger and threatened the visiting nurses by “*venting their anger about unfairness of the visiting nurse.*” In addition, given that home visits occur in the clients’ space, not only verbal sexual harassment but also non-verbal sexual stimulation could occur. For example, visiting nurses complained that, during home visits, the clients often wore revealing clothes or would cause discomfort with audio or visual pornography.

Although visiting nurses checked the patient information before the visit, insufficient and inaccurate information could lead to exposure of violent situations: *“I thought this man was living alone, but one time, his son suddenly followed me out with a bottle” (Participant 5).* There were also cases of violence from the patients’ neighbors.

If a personal phone number was accidently disclosed to the client during work, the bullies often continuously called the nurse *“like a stalker”.* There were also cases where a client came to the visiting nurse’s office after the visit and harassed her.

#### 3.3.2. Changes after Experiencing Violence

Visiting nurses who experienced workplace violence during home visits were worried and even afraid. In particular, when the nurse experienced violence from a mentally ill patient, they often refused to visit other patients with mental illness. They also complained of fear, which was followed by lower self-esteem and more negative thoughts.

The visiting nurses confessed that they were hesitant to provide services or became passive after experiencing violence. *“After a single experience, you don’t want to work anymore. Why do I have to do this while enduring such treatment? It’s a big barrier. But no one can solve this problem for me” (Participant 3).*

#### 3.3.3. Individual Efforts to Respond

Visiting nurses said that they could “*roughly screen*” high-risk clients. *“Since I’ve been working here for so long, what can I say, I know just from hearing their voice” (Participant 2)*. When they detected any risk, they prepared themselves to prevent any violent situations (e.g., offering health counseling outside of the house if the home environment was considered dangerous). More experienced visiting nurses were prepared to cope with violence with their accumulated know-how and years of experience. Therefore, experienced visiting nurses could cope with any situations relatively well, but novices were relatively inexperienced dealing with such situations. *“Old and experienced nurses like me should come out and say, ‘Oh, sir, I’ll come next time because your outfit is uncomfortable for us to consult.’ But how embarrassing would young nurses be in the same situation?” (Participant 4).*

Visiting nurses struggled between duty and safety, which emphasized *“trust”* with the client while trying to *“avoid exposure”* of personal information at work to prevent harassment that could persist after the home visit.

#### 3.3.4. Poor Organizational Management

Visiting nurses who participated in the FGI described a lack of systematic management of workplace violence during home visits. Prior to the visit, it was difficult to obtain detailed information about the clients, especially whether they had a criminal history including sexual harassment or mental illness. In addition, the visiting nurses felt that it was not beneficial to be given self-defense tools and that the support was lacking. Many visiting nurses reported that they went on home visits without enough training on workplace violence. *“I think we need education on how to deal with workplace violence, and I hope there are opportunities to listen to some of the experienced visiting nurses” (Participant 4).*


Strategies for preventing violence have been implemented in each district in Korea but with considerable variation. In some districts, home visits were conducted in teams for safety (two persons per team). However, in most districts, this was not the case. In one district, the visiting nurses shared their schedules with other team members, while, in other districts, the visiting nurses did not know each other’s routes. There was also a difference in the prevalence of smart watches that were provided in case the nurses needed to request emergency assistance due to violence during a home visit. Some visiting nurses also said that, despite the availability of smart watches, they were less useful in their work.

Others pointed out that there was a lack of a supportive culture in which managers and other staff members cooperated and cared about the safety of the visiting nurses. When they were asked to accompany the visiting nurses for protection, supervisors *“wouldn’t even listen and said they don’t have time…There is no sick leave or leave to recover (even in the event of violence). Then the visiting nurse will have to visit again to the area the next day. There are no countermeasures, so I felt it was very difficult to work again without any action after the trauma” (Participant 4).* In addition, since visiting nurses were often hired on a temporary basis, there was a problem when the visiting nurses changed frequently. The nurses reported that there was no continuity of work and *“such situations could happen again*” in the home of the client where the incident occurred.

## 4. Discussion

### 4.1. Status of Workplace Violence

The present study demonstrated that 75.8% of the visiting nurses had experienced violence, and 67.2% of these nurses had experienced violence within the past year. Compared to previous studies [4,5,6,9], the proportion of violent experiences was similar or slightly higher in this study. In addition, the proportion of verbal violence was the highest among all types of violence [4,5]. Nurses who participated in the FGI said that, when clients are dissatisfied with their welfare benefits, they often verbally expressed their dissatisfaction and anger to the visiting nurses. Verbal violence is important not only because it occurs more frequently than other forms of workplace violence, but especially because it causes emotional damage in the long run [33]. The nurses’ perception was that verbal violence is unavoidable due to the nature of face-to-face, in-home work and, thus, they may be reluctant to report verbal violence [23]. It is necessary for supervisors and trainers to warn visiting nurses that the position presents unique problems beyond usual working environments.

Registered nurses (RNs) in clinical practice have reported that they experience higher levels of physical violence than sexual violence [34,35]. However, visiting nurses in this study experienced higher rates of sexual violence than physical violence. This is understandable given that the place where the work is performed is the patient’s private home [4]. In particular, the nurses reported that sexual violence during home visits occurred not only verbally but also in nonverbal ways, which could lead to sexual shame for the nurses.

We also found that visiting nurses may be at risk of violence due to the environment of the client’s surroundings. Galinsky et al. reported that if home healthcare workers perceive a threat of violence by others in and around the patients’ homes, they are at greater risk of violence during home visits [12]. Thus, it is important to pay attention to the risk factors that visiting nurses recognize and report.

When visiting homes and providing services, workers must rely on their own resources rather than the protection of the organization [5]. Home-visiting personnel should develop the capacity to assess potential risks and cope with violent situations, which should be provided as an organizational education program [20]. However, our findings revealed that visiting nurses are exposed to violence during home visits without systematic education and training. In this study, about half (46.5%) answered that they were not given any educational and behavioral guidelines to cope with the violence in the homes of visiting clients. Visiting nurses complained about the lack of education or training in workplace violence and wanted more tools and education.

As shown in the focus group interviews, the visiting nurses prevented some workplace violence based on their personal experience and know-how accumulated from their years of service rather than based on systematic education and training. In this study, there was a significant correlation between the age and experience of visiting nurses (r = 0.37, *p* < 0.001). As the age increased, the risk of workplace violence decreased significantly (OR = 0.91, 95% CI = 0.86–0.95). In general, a higher proportion of workplace violence experiences were reported among the younger nurses [36,37]. The reasons for this can be interpreted in many ways, but one interpretation is that younger people may lack the experience of coping with violent situations [10]. In this study, it is expected that the professional attitudes of visiting nurses who were older and had longer careers had the effect of suppressing violent incidents. In particular, if the *“feeling”* was not in safe, the risk of experiencing violence may have been reduced because the visiting nurse was prepared for the risk of violence before the visit. However, organizations should provide systematic education and training programs rather than relying on individual sensibilities and experience.

### 4.2. Recommendations and Suggestions

The home visit process can be divided into pre-visit, during-visit, and post-visit. We found that workplace violence experienced by visiting nurses did not only occur when they visited the home but could also occur before and after the visit. Therefore, organizational management of workplace violence should be addressed at each stage.

In the pre-visit stage, it is important to screen and prepare for risk factors. In terms of the preventive aspects of the safety management system in this study, the visiting nurses recognized that there was a lack of a systematic safety management system to prevent violence from patients or family members. This was made more evident in the FGI, where visiting nurses described overcoming the limitations of poor organizational management with their own personal efforts.

The use of risk assessment tools, including the organization’s preventive policies, is an important strategy to strengthen the safety of home visits [3]. Various risk factors require prior screening, but it is worth paying attention to patients’ history of violence against a visiting nurse or criminal history including sexual violence, as identified in this study. For a client who has previously been violent during a visit, there is a risk of additional violence if the visiting nurse revisits without prior screening and prior appropriate action being taken. Visiting nurses who participated in the FGI pointed out that, if the visiting nurse in charge of the district changed frequently due to a temporary work contract, the record was likely to be lost and the problem was likely to be repeated. In addition, the results showed that the risk of sexual violence increased when visiting nurses were exposed to clients with a history of sexual violence. However, the visiting nurses who participated in the FGI complained that it was difficult to gain access to clients’ criminal history in advance. They were screening clients by “feeling” due to the lack of prior information. Visiting nurses could check only the presence or absence of mental illness in the Public Health Information System and collect medical diagnosis and medication information from clients at the visit, which makes it difficult to identify the patient’s condition before the visit. Focus group interviews showed that, even if basic information about the client were available, nurses could be exposed to unexpected danger. Providing accurate information about clients and keeping records could help ensure the safety of visiting nurses. Based on prior information, a protocol should be established to screen the risks of home visits through screening tools, and to discuss the risk of expected workplace violence with the manager so nurses could be assigned in pairs and other necessary precautions for potentially dangerous clients (e.g., accompanying police) [19,23].

Several unique factors impact the support system of visiting nurses. Many community visiting nurses leave the office without informing anyone of where they are going and when they will return. In addition, they work without a response plan if they do not return to the office after a scheduled visit or a plan for how to call for help if needed [8]. Visiting nurses also often work alone, which makes them more susceptible to unsafe events [7]. In a study by Fujimoto et al. [24], managing visit schedules and identifying the location of visiting personnel during a visit were negatively related to workplace violence exposure. Thus, a stronger support system is needed to track the visit routes and provide immediate assistance in the event of an emergency. Such a safety monitoring system could provide added stability for visiting nurses [23].

We noted that harassment can persist after a visit, according to the FGI results related to the nurses’ workplace violence experience. The visiting nurses who participated in the interviews said that harassment continued through text messages or phone calls when an individual’s mobile phone number was accidently disclosed during work. Visiting nurses struggled between duty and safety and tried hard not to disclose personal information because such repeated and undesired communication, or stalking, caused considerable suffering for the visiting nurses. These findings are consistent with previous reports that healthcare providers are generally at higher risk for stalking [38,39,40]. This is because those who receive the service have an attachment and expectations about their relationship with the provider, and, if those expectations are not met, anger and revenge can be directed at the healthcare provider [38,40].

The interviewed visiting nurses said that they were worried, fearful, and had negative feelings after a violent incident. Fear is one of the most common consequences of healthcare workers who experience workplace violence [41,42]. Fernandes et al. found that this fear could be generalized not only to those who were the perpetrators of violence, but also to those who thought they were potentially viable [42]. The visiting nurses also said that their attitudes toward visiting services changed after experiencing violence. Several previous studies have pointed out that, if a visiting nurse is exposed to violence, it may affect the provision of services as well as the response by individual visiting nurses [6,18,21,43]. Nurses who have been exposed to violence may reduce the amount of time they spend in their clients’ homes or avoid visits by themselves, which may affect the delivery of nursing services [6]. A violent experience may also have a negative impact on the nurse’s interaction with patients if the nurses are anxious, depressed, or less motivated after an attack [6]. Thus, in the event of a violent incident, early intervention is needed to minimize the impact.

However, looking at the response to post-event management, it can be expected that, when a violent incident occurs during a home visit, nurses are often treated as being helped or comforted by a supervisor or co-worker rather than having a formal problem-solving procedure. Administrative measures, such as professional counseling and work coordination for emotional support, and workers’ compensation should be provided if necessary.

### 4.3. Limitations of the Study

The present study has several limitations. First, since the quantitative study used a voluntary retrospective survey and was conducted in an open space of the education center, it is likely that the actual workplace violence incidences were under-reported. Second, some of the risk factors investigated in this study had few events, so the confidence intervals were widely estimated. Third, although the reliability for some of the scales in terms of Alpha was rather modest, this critique does not apply to constructs like risk factors because risk factors refer to a formative rather than a reflective construct. They cannot be expected to be highly correlated among each other [44]. Fourth, all recruited participants for the FGI had more than 10 years of experience as visiting nurses. Younger nurses with less experience may have had different perspectives. Lastly, this study is somewhat descriptive, so more advanced studies need to be conducted in the next stage. In this study, visiting nurses’ recognition of a safety management system was confirmed, but it is necessary to confirm the actual impact on the organization’s safety management system for each type of violence. Although an FGI was conducted in this study, a more in-depth approach such as a case study or individual interview is needed in the future.

Nevertheless, this study raises important issues for further research, such as the need to investigate each violent incident by considering the degree and intensity of the violent incident, developing a screening tool to evaluate the risk factors identified in this study, developing workplace violence prevention programs, and further evaluating visiting nurses’ experiences.

## 5. Conclusions

This study was conducted using a mixed research method to identify the workplace violence experienced by visiting nurses during home visits. The quantitative data identified the types of violence experienced by visiting nurses, the perceived workplace risk factors, and nurses’ perceptions of the safety management system. Workplace violence was reported by 75.8% of the nurses over their entire career as a visiting nurse, and 67.2% experienced workplace violence within the previous year. In addition, logistic regression analysis identified risk factors affecting visiting nurses’ workplace violence experience. The probability of workplace violence experience decreased for older nurses. Compared to permanent workers, temporary workers had an increased prevalence of workplace violence. Clients who had been violent toward visiting nurses in the past showed an increased prevalence of continued violence. The FGI also revealed the nature of the workplace violence the visiting nurses faced and the changes in nurses’ attitudes including fear after experiencing violence. We also found various aspects of visiting nurses’ individual efforts when facing poor organizational management. Based on the results of this study, we identified evidence of poor workplace violence management plans, according to the spatial and process characteristics of visiting nursing work. To keep up with the growing demand for community-based health care [1,2], it is urgent to establish a protection system for visiting nurses. It also requires attention and compensation for health problems that can occur after home visiting nurses experience workplace violence.

## Figures and Tables

**Table 1 ijerph-17-04222-t001:** Characteristics of visiting nurses.

Variables	Categories	Survey Respondents (*N* = 357)	Focus Group Interview Participants (*N* = 6)
		*n* (%) or Mean ± SD	*n* (%) or Mean ± SD
Gender	Male	3 (0.8)	0 (0.0)
	Female	354 (99.2)	6 (100.0)
Age (year)		50.15 ± 6.75	56.67 ± 4.38
	≤29	4 (1.1)	0 (0.0)
	30–39	17 (4.8)	0 (0.0)
	40–49	136 (38.4)	1 (16.7)
	50–59	172 (48.6)	3 (50.0)
	≥60	25 (7.1)	2 (33.3)
	Missing	3	
Marital status	Married	335 (94.1)	6 (100.0)
	Single ^1^	21 (5.9)	0 (0.0)
	Missing	1	
Level of education	ADN	193 (54.4)	3 (50.0)
	BSN	153 (43.1)	3 (50.0)
	≥MSN	9 (2.5)	0 (0.0)
	Missing	2	
W/E as RN (month)		174.46 ± 71.48	216.17 ± 32.59
	Missing	17	
W/E as visiting nurse (month)		68.46 ± 43.74	137.00 ± 11.76
	Missing	2	
Types of employment	Permanent	310 (87.6)	3 (50.0)
	Temporary	44 (12.4)	3 (50.0)
	Missing	3	
Number of patients		460.51 ± 202.03	-
	Missing	12	
Number of visits (month)		81.51 ± 28.73	-
	Missing	20	
Appearance (duplicate response)	Uniform only	73 (20.4)	-
	ID card only	255 (71.4)	
	Casual clothes and no ID card	58 (16.2)	
	Other	27 (7.6)	

^1^ Widowed, divorced, unmarried. ADN = associate of arts degree in nursing, BSN = bachelor of science in nursing, MSN = master of science in nursing, W/E = work experience, and RN = registered nurse.

**Table 2 ijerph-17-04222-t002:** Workplace violence experience (*N* = 357).

Variables	Categories	*n* (%)
Workplace violence experience for the whole working period (*n* = 356)	Yes	270 (75.8)
	No	86 (24.2)
Workplace violence experience within the past 1 year	Yes	240 (67.2)
	No	117 (32.8)
K-WVS: exposure and experience of violence ^1^	Mean ± SD	1.47 ± 1.41
Verbal violence	Yes	191 (53.5)
	No	166 (46.5)
Sexual violence	Yes	108 (30.3)
	No	249 (69.7)
Threats or harassment	Yes	100 (28.0)
	No	257 (72.0)
Discrimination	Yes	110 (30.9)
	No	246 (69.1)
Physical assault	Yes	8 (2.2)
	No	349 (97.8)

^1^ K-WVS = Korean Workplace Violence Scale (possible range: 0–15). The instrument includes five questions of verbal violence, sexual violence, threat or harassment, discrimination, and physical assault.

**Table 3 ijerph-17-04222-t003:** Percentage reporting workplace risk factor exposure within one year (*N* = 357).

Variables	Percentage		Categories ^1^	
	Mean (SD)	Median (IQR)	Category	*n* (%)
Hazardous residential environment (*n* = 354)	33.6% (27.2)	30.0% (10.0–50.0)	>0%	337 (95.2)
			0%	17 (4.8)
Working alone (*n* = 353)	62.1% (34.9)	80.0% (30.0–90.0)	>0%	325 (92.1)
			0%	28 (7.9)
Working with HRC: Psychiatric disorder (*n* = 339)	17.3% (18.4)	10.0% (5.0–30.0)	>0%	332 (97.9)
			0%	7 (2.1)
Working with HRC: History of a suicide attempt (*n* = 227)	6.9% (9.5)	5.0% (1.0–10.0)	>0%	252 (91.0)
			0%	25 (9.0)
Working with HRC: Alcoholic (*n* = 317)	12.1% (14.8)	10.0% (2.0–20.0)	>0%	306 (96.5)
			0%	11(3.5)
Working with HRC: Domestic violence assailant (*n* = 222)	5.3% (7.5)	2.0% (0.1–10.0)	>0%	173 (77.9)
			0%	49 (22.1)
Working with HRC: Child abuser (*n* = 181)	2.3% (4.1)	0.0% (0.0–3.0)	>0%	84 (46.4)
			0%	97 (53.6)
Working with HRC: Criminal record (*n* = 235)	8.1% (12.8)	5.0% (1.0–10.0)	>0%	197 (83.8)
			0%	38 (16.2)
Working with HRC: Sex offender (*n* = 317)	3.3% (10.2)	0.0% (0.0–2.0)	>0%	144 (45.4)
			0%	173 (54.6)
Working with HRC: History of violence against visiting nurses (*n* = 320)	2.8% (6.7)	0.0% (0.0–2.0)	>0%	134 (41.9)
			0%	186 (58.1)
Working with HRC: In distress (*n* = 348)	25.5% (25.5)	20.0% (5.0–40.0)	>0%	336 (96.6)
			0%	12 (3.4)

^1^ Risk factors measured with continuous variables (%) were divided into binary variables: IQR = interquartile ranges, and HRC = high-risk clients.

**Table 4 ijerph-17-04222-t004:** Visiting nurses’ perception of the safety management system (*N* = 357).

Domain	Question	Categories	*n* (%)
Preventive measure (*n* = 342)	In the workplace, there is a device or system to protect visiting nurses from violence from patients or their family members.	Yes	112 (32.6)
		No	232 (67.4)
	Workplaces take a variety of measures to prevent violence from visited patients or their family members.	Yes	119 (34.6)
		No	225 (65.4)
	There are educational programs and behavioral guidelines in the workplace to cope with the violence from visited patients or their family members.	Yes	184 (53.5)
		No	160 (46.5)
	Workplaces are safe to work and provide protection from violence from patients or their family members.	Yes	119 (34.6)
		No	225 (65.4)
	In the event of violence from patients or their family members, there is a process to identify the incident (instructions, internal regulations, etc.) in the workplace.	Yes	92 (26.7)
		No	252 (73.3)
Post-event management (*n* = 345)	I have a supervisor who helps me solve the problem when I have been exposed to violence.	Yes	153 (44.7)
		No	189 (55.3)
	I have a colleague who helps me solve the problem when I have been exposed to violence.	Yes	196 (57.3)
		No	146 (42.7)
	My organization sympathizes with and consoles the heartache of being assaulted.	Yes	186 (54.4)
		No	156 (45.6)
	My organization solves problems caused by violence.	Yes	113 (33.0)
		No	229 (67.0)

**Table 5 ijerph-17-04222-t005:** Effect of risk factors and protection policy on workplace violence experience.

Variables	Categories	Total	Verbal	Sexual	Threat/ Harassment	Discrimination
		OR (95% CI)	OR (95% CI)	OR (95% CI)	OR (95% CI)	OR (95% CI)
**Individual Risk Factors**						
Age (year)		**0.91** **(0.86–0.95)**	**0.91** **(0.87–0.95)**	**0.96** **(0.91–1.00)**	**0.95** **(0.90–0.99)**	**0.94** **(0.90–0.99)**
Types of Employment	Permanent	1.00	1.00	1.00	1.00	1.00
	Temporary	**2.95** **(1.10–7.88)**	**2.66** **(1.05–6.72)**	1.56 (0.65–3.71)	**3.57** **(1.44–8.81)**	**3.18** **(1.30–7.78)**
Number of registered patients (100 persons)		1.06 (0.92–1.21)	**1.17** **(1.01–1.35)**	0.95 (0.83–1.09)	1.11 (0.97–1.27)	0.99 (0.86–1.13)
**Workplace Risk Factors**						
Working with HRC: Child abuser	Yes	1.35 (0.70–2.60)	1.18 (0.61–2.27)	0.83 (0.34–2.03)	1.31 (0.58–2.97)	**2.75** **(1.49–5.09)**
	No	1.00	1.00	1.00	1.00	1.00
Working with HRC: History of violence against visiting nurses	Yes	**3.76** **(1.97–7.15)**	**4.97** **(2.73–9.06)**	**2.22** **(1.26–3.91)**	**3.78** **(2.06–6.92)**	**1.91** **(1.02–3.59)**
	No	1.00	1.00	1.00	1.00	1.00
Working with HRC: Sex offender	Yes	1.83 (0.87–3.85)	1.51 (0.84–2.72)	**2.90** **(1.49–5.65)**	1.57 (0.84–2.92)	1.14 (0.56–2.30)
	No	1.00	1.00	1.00	1.00	1.00

Note. For input variables, only significant results are shown in this table. Table A3 reports the overall logistic regression results: gender, age, education level, marital status, type of employment, number of registered patients, number of visits per month, appearance, hazardous residential environment, working alone, working with HRC: psychiatric disorders, history of a suicide attempt, alcoholic, domestic violence assailant, child abuser, criminal record, sex offender, history of violence against visiting nurses, people in distress, and prevention policy. HRC = high-risk clients. OR = odds ratio. CI = confidence interval. Bold font indicates statistical significance.

**Table 6 ijerph-17-04222-t006:** Outcome of qualitative analysis.

Categories	Themes	Subthemes
Violence experienced and the consequences	Violence visiting nurses faced	Verbal expression of dissatisfaction and anger
		Stimulation of sexual shame
		Unpredicted danger
		Harassment continued after the visit
	Changes after experiencing violence	Increases fear
		Expanded negative emotions
		Changes in attitudes toward visiting-related services
Reality of violence management	Individual efforts to respond	Screening by feeling/emotional response
		Act/cope with years of experience
		Efforts to avoid disclosure of personal information
		Struggle between duty and safety
	Poor organizational management	Lack of prior information and support
		Inadequate education on coping
		Deviation from the prevention strategies
		Lack of cooperation and understanding from other occupations
		Lack of continuity in work

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
