# Peer review of "Who Cares for Visiting Nurses? Workplace Violence against Home Visiting Nurses from Public Health Centers in Korea"

_ijerph, 2020, doi:10.3390/ijerph17124222_

Round 1

Reviewer 1 Report

The authors answered most of my concerns and made some useful changes to the manuscript which has been relevantly improved.

Therefore, I do not have further comments or suggestions for Authors.

Author Response

No further comments. Thank you.

Reviewer 2 Report

I have taken a look at the revision and your replies to my comments, now. I think you have done a good job discussing most of the issue I have raised in my last review.

  1. In my view, the way you present the motivation of the study has not improved considerably. The focus continues to appear quite arbitrary to me. I recommend stating clearly which question or which problem is addressed and how the current study is derived from prior research.
  2. Thank you for providing the correlations in the appendix. I think you should provide the full correlation matrix (all variables x all variables). For instance, the safety management perceptions are currently absent from the correlation table.
  3. I appreciate that you included a few words on the low alphas of some of the measures. However, my impression is that your discussion regarding formative vs. reflective constructs is misleading. I recommend taking a look at the paper by MacCallum I mentioned in my last review. The bottom line is that alpha is not a valid measure of reliability for formative constructs. Please revise the paragraph.
  4. Admittedly, my impression is that your response to my comment regarding the factor structure of the safety management scale basically restated the justification that is already in the manuscript. I propose running a factor analysis to make sure the empirical structure is in line with the advice from the expert. Given that you have modified (shortened) an existing scale you should provide evidence that is reliable and valid. Merely stating that items 1-4 measure construct and items 5-10 measure construct B is never very convincing. You do have the opportunity to provide evidence for the reliability of the scale applying your dataset. I propose making use of the data here.

Author Response

Attached are the responses for your review comments. Thank you. 

Reviewer 3 Report

Please add a researcher section so you tell us more about you as the researcher(s) and your connection to this study (align with personal interests, professional work, etc.).

Describe the recruitment process and demographics for the six participants in the focus group interview. Please share the questions in the interview guide you used in your study. Were they open-ended? Did you give participants an opportunity to provide to provide follow-up information or share their own experiences and perspectives?

Author Response

(The authors gave the same response as above.)

Reviewer 4 Report

Dear Authors,

thank your for revising your manuscript according to my comments. After revising the study's objective, results now meet the expectations of the readers.

I agree with you that further knowledge is needed on the current phenomenon of workplace violence against visiting nurses, in order to understand what can be the best practices and procedures to be applied at OSH management level. This might have also an impact at policy level. 

Your study offers a good description of the home visiting nurses’ experiences of workplace violence and visiting nurses’ perceptions of the potential risk factors and organizational management of workplace violence already put in place (or not as mainly happened).

After revisions, the descriptive nature of the study is now clear and there are not expectations of identifying the role of OSH management practices in workplace violence.

As you identified, research in this field is quite lacking, thus I recognize the value added of having a descripting study having both quantitative and qualitative data. Also implications for practices are clearer now.

Author Response

No further comments. Thank you. 

Round 2

Reviewer 2 Report

I think you addressed my concerns adequately. I disagree with a couple of specific points made in the paper, but I think you convey the contribution of the manuscript better than in prior versions.

I appreciate the effort that went into the documentation of the scales and variables. I think the factor analysis supports your decision to group the items to the respective factors.

I appreciate your revision of the discussion regarding the low reliabilities. The following sentence is not clear or at least it confuses me:

For the convergent validity of the formative measurement, correlation analysis with reflective latent variables would be necessary. However, we did not include this data collection at the design stage of the study, so validity assessment cannot be conducted.

One suggestion: Omit this sentence and write something like. Although the reliability for some/most of the scales in terms of Alpha was rather modest, this critique does not apply to constructs like risk factors, because risk factors refer to a formative rather than a reflective construct. That is, they cannot be expected to be highly correlated among each other (see MacCallum et al.).

On a related note: The use of a formative construct per se is not a limitation of the study*. The modest reliabilities may be considered a limitation, but you can explain that your measurement is probably fine at least for formative constructs. Your sentence above refers to validity. I think, no one would expect you to provide evidence of validity of a validated scale (unless you introduce it for the very first time).

To cut a long story short: Admit the modest reliabilities and argue that for most of the scales measurement is probably fine.

I am sorry to be nagging about this issue again. But I think it should be stated correctly.

Final point: I appreciate the inclusion of the full correlation table. Could you please add the names of the variables in the first column (instead of X1 etc.)? This will increase the reader-friendliness of the table considerably.

Thank you in advance.

Author Response

Dear Editor,

Thank you very much for having considered our manuscript titled “Who cares for visiting nurses? - Workplace Violence against Home Visiting Nurses from Public Health Centers in Korea.”

We want to thank the editor and reviewers for their critiques and revision recommendations. Our detailed, point-by-point responses to the reviewer comments are given below, whereas the corresponding revisions are marked in colored text in the manuscript file. Specifically, green text indicates the 3rd changes, red text indicates the 2nd, and blue text indicates changes made according to the 1st comments.

We would like to thank you once again for your consideration of our work and inviting us to submit the revised manuscript. We look forward to hearing from you.

#1 Comment

I think you addressed my concerns adequately. I disagree with a couple of specific points made in the paper, but I think you convey the contribution of the manuscript better than in prior versions.

I appreciate the effort that went into the documentation of the scales and variables. I think the factor analysis supports your decision to group the items to the respective factors.

I appreciate your revision of the discussion regarding the low reliabilities. The following sentence is not clear or at least it confuses me:

For the convergent validity of the formative measurement, correlation analysis with reflective latent variables would be necessary. However, we did not include this data collection at the design stage of the study, so validity assessment cannot be conducted.

One suggestion: Omit this sentence and write something like. Although the reliability for some/most of the scales in terms of Alpha was rather modest, this critique does not apply to constructs like risk factors, because risk factors refer to a formative rather than a reflective construct. That is, they cannot be expected to be highly correlated among each other (see MacCallum et al.).

On a related note: The use of a formative construct per se is not a limitation of the study*. The modest reliabilities may be considered a limitation, but you can explain that your measurement is probably fine at least for formative constructs. Your sentence above refers to validity. I think, no one would expect you to provide evidence of validity of a validated scale (unless you introduce it for the very first time).

To cut a long story short: Admit the modest reliabilities and argue that for most of the scales measurement is probably fine.

-> Following your advice, we have revised the paragraph [4.3. Limitations of the study, lines 462-465].

#2 Comment

Final point: I appreciate the inclusion of the full correlation table. Could you please add the names of the variables in the first column (instead of X1 etc.)? This will increase the reader-friendliness of the table considerably. Thank you in advance.

-> Following your advice, we added the names of the variables in the first column. See Appendix B (Table B1). 

This manuscript is a resubmission of an earlier submission. The following is a list of the peer review reports and author responses from that submission.

Round 1

Reviewer 1 Report

The paper is solid and well thought-out. It addresses an interesting and emerging topic and certainly advances our understanding of workplace violence against home visiting nurses.

However, I have a couple of suggestions:

1)    Introduction, line 62: in addition to reference [3], it would be useful to cite also 2 more recent papers on this specific topic, i.e.

·         Experiences of Violence and Preventive Measures Among Nurses in Psychiatric and Non-Psychiatric Home Visit Nursing Services in Japan. Fujimoto H, Greiner C, Hirota M, Yamaguchi Y, Ryuno H, Hashimoto T. J Psychosoc Nurs Ment Health Serv. 2019 Apr 1;57(4):40-48. doi: 10.3928/02793695-20181023-04.

·         Violence exposure and resulting psychological effects suffered by psychiatric visiting nurses in Japan.Fujimoto H, Hirota M, Kodama T, Greiner C, Hashimoto T. J Psychiatr Ment Health Nurs. 2017 Oct;24(8):638-647. doi: 10.1111/jpm.12412.

2)    Discussion, paragraph 4.3. Limitations of the study, line 402: in addition to the 3 limitations mentioned by the Authors, I do think that it should also be noted that the study is based only on subjective data while objective data are missing (e.g. clinically diagnosed psychiatric disorders and/or use/abuse of psychoactive drugs among home visiting nurses).

3)    Conclusions, line 424: I would suggest to the Authors to add also a consideration about the need of more recognition and compensation for work-related stress disorders among home visiting nurses.

Reviewer 2 Report

I have read the manuscript entitled „Who cares for visiting nurses? - Workplace Violence against Home Visiting Nurses from Public Health Centers in Korea“ with great interest. I think incivility and violence against professionals in the healtcare sector is a topical issue which has received considerable media coverage. Below, I outline a couple of issues that came to my mind upon reading the paper. I hope they will be helpful for going on with your research. 1. Theory 1.1. While I think you manuscript taps into a very relevant issue, I missed a clear rationale of what the actual contribution tot he literature is. I understand that research on violence against medical personnel is probably in a nascent state, but from the introduction it did not become clear to me what we already know, what we don’t know, and how your study contributes to fill the gap in the literature and why that is important. I recommend stating more explicitly and clearly what the unique contribution of your paper is. 1.2. I appreciate the mixed methods approach, because it eventually provides a rich account particularly of underresearched topics. However, I must say that I have a hard time agreeing with the order of doing *quantitative“ research first. Wouldn’t one normally start with *qualitative* studies to get an overview of relevant aspects and continue with quantitative research? 1.3. On a related note, your review of the literature and use of specific scales suggests that there is quite a bit research on violence in healthcare. Hence, pointing out the unique contribution of your study is crucial. Of note, exploring or transferring an issue that has been researched more or less extensively in a different domain to another profession per se is not a contribution in my view. In other words, I suggest that you explain the specific and unique contribution of your study more explicitly. This might also require a more thorough review of the literature. Rather than stating that we have little knowledge of the phenomenon, I should report evidence from prior research and leverage these findings to make a case for doing your study. 2. Methods 2.1. I think dichotomization of theoretically continuous variables needs explicit justification. I understand that the frequency measures are probably highly skewed. However, you throw away (possibly relevant) variance in your measures and may even draw wrong conclusion. See, for instance, the discussion on the practice of median split [1]. For some of the measures the cut-off value was not justified. To avoid confusion, I do not intend to suggest abandoning dichotomization. However, I urge you to be explicit about the rationale for dichotomizing and the issues associated with dichotomization. Of note, you should apply the set of criteria consistently acorss variables. You might report correlations among continuous variable scores in addition to the logistic regression results. If results converge, this would strengthen your conclusion about the relevant predictors of violence. 2.2. Minor issue: The alpha value of the violence scale is quite modest. This is probably because violence is a formative rather than a reflective construct [2]. You may want to discuss this issue to qualify the reliability of your measures. They are probably fine. 2.3. It did not become clear to me why you choose 5 participants for the focus group interviews only. Being no expert in qualitative research, my impression was that the sample size is quite low. On might argue that 5 participants may not be representative of the population being exposed to violence. I know, there are criteria for determining a point where additional interviews do not provide much additional insights. I suggest justifying the sample size explicitly. 2.4. My impression was that one of the preventive measure scale items (the last one) may correspond better to post-event management, because it refers to activities relevant after an incident of violence. Given that safety management is probably a reflective construct, you could apply factor analysis to identifiy the structure empirically. I do not want to arge you to ran factor analysis, but you should make a strong case for why and how you distinguish specific aspects of safety management. At a minimum, you should state clearly what defines post-event vs. Preventive measures. It should not look arbitrary. 2.5. I might have missed that but did you provide an overview of the interview guide. I think, presenting the questions or themes of the questions would be quite helpful. You might provide it as a supplementary material. 3. Results 3.1. Ideally, you should also provide the correlation matrix for the focal variables. 4. Discussion 4.1. In the implications section you formulate quite specific practical implications regarding intervention and prevention. This section highlights one potential contribution of your paper. You may consider stating the issue of identifiying leverage points for intervention in the introduction. 4.2. I think your results are mainly descriptive. Although, purely descriptive research is seen very critically in my domain of IO-psychology, I understand that IJERPH is an interdisciplinary journal and that decriptive research is more typical in public health research. However, it is not clear how representative your results are for nurses in Korea, Asia or around the globe. I think, this is a major weakness of your study, which limits the overall scientific contribution of your study. At a minimum you should discuss this issue as a limitation of your research. 5. Style 5.1. Although, I have a couple of concerns regarding the content and analyses, I think you have provided a very diligently prepared manuscript here, which is easy to read, and fine regarding language and style. 6. Reference 1. MacCallum, R.C.; Zhang, S.; Preacher, K.J.; Rucker, D.D. On the practice of dichotomization of quantitative variables. Psychol. Methods 2002, 7, 19–40. 2. Ellwart, T.; Konradt, U. Formative versus reflective measurement: An illustration using work–family balance. J. Psychol. Interdiscip. Appl. 2011, 145, 391–417.

Reviewer 3 Report

Provide information on the background and intent of the survey, generalizability, and reliability and validity data. Also, include more detail on the target population and discussion should include gender and age.  Address if the 532 questionnaires were all or part of the total in-service nurses.  

Provide the rationale for the six participants in the interview and greater detail on the interview protocol. To support this was not a focus group and rather a group interview, please give further detail on the role of the interviewer and the interview guide. There may be value in adding add a section about the researcher(s) and the connection to this study.

The conclusion should be supported with the literature and referenced to findings.

Grammar:  Please check use of pronouns and consistency in tense. 

Reviewer 4 Report

Dear Authors,

I was pleased of having the occasion to read this manuscript. The topic of Violence and Harrassment at work is emerging, particularly in healthcare and jobs concerning isolation and relationships with clients. Thus, I recognized the importance of improving evidence in this field, particularly in the light of informing interventions in workplace practice and procedures to deal with this issue.

Unfortunately, the study analysis and results did not account sufficiently of the most important aspect under investigation that is identify the organizational management of workplace in the light of understanding its role on violence episodes and what must be improved. .

The purpose reported by authors is “The purpose of this study was to explore the home visiting nurses’ experiences of workplace violence, and to identify the risk factors and organizational management of workplace violence.

However, only the effect of risk factors on workplace violence experience is investigated by logistic regressions. At this regard information of an effect of demographics (age, gender, contract..) and working with high risk client is not sufficient to understand how to intervene to avoid violence episodes in nurses visiting. Reading the manuscript I was expected some findings accounting the effects of policy procedures or safety management system on violence episodes to verify (some direct link or moderation effect on the relationship between risks and episodes). Even if quantitative data on safety management system were collected, authors limit to report descriptives about this and this lowers the potential of their study findings.

I think the study design must include a broader quantitative investigation about this to fully answer to the purpose of the study. Moreover, the discussion should be improved accordingly, offering practical solutions and future perspectives useful for workplaces.

My suggestion is that authors improve their analysis and results in this direction, to have a study really offering a practical contribution in this field, and then resubmitting the manuscript to the journal.